# Supplementation of Enriched Polyunsaturated Fatty Acids and CLA Cheese on High Fat Diet: Effects on Lipid Metabolism and Fat Profile

**DOI:** 10.3390/foods11030398

**Published:** 2022-01-29

**Authors:** Monica Tognocchi, Maria Conte, Lara Testai, Morena Martucci, Andrea Serra, Stefano Salvioli, Vincenzo Calderone, Marcello Mele, Giuseppe Conte

**Affiliations:** 1Department of Agriculture, Food and Environment, University of Pisa, Via del Borghetto, 80, 56124 Pisa, Italy; moni.tognocchi@gmail.com (M.T.); andrea.serra@unipi.it (A.S.); marcello.mele@unipi.it (M.M.); 2Dipartimento di Medicina Specialistica, Diagnostica e Sperimentale, University of Bologna, 40126 Bologna, Italy; m.conte@unibo.it (M.C.); morena.martucci3@unibo.it (M.M.); stefano.salvioli@unibo.it (S.S.); 3Dipartimento di Farmacia, University of Pisa, 56126 Pisa, Italy; lara.testai@unipi.it (L.T.); vincenzo.calderone@unipi.it (V.C.); 4Research Center of Nutraceutical and Food for Health, University of Pisa, Via del Borghetto, 80, 56124 Pisa, Italy

**Keywords:** cheese, high-fat diet, lipid metabolism, CLA, PUFA ω-3, inflammation, obesity, metabolic syndrome

## Abstract

Epidemiological studies have demonstrated a positive relationship between dietary fat intake and the onset of several metabolic diseases. This association is particularly evident in a diet rich in saturated fatty acids, typical of animal foods, such as dairy products. However, these foods are the main source of fatty acids with a proven nutraceutical effect, such as the ω-3 fatty acid α-linolenic acid (ALA) and the conjugated linoleic acid (CLA), which have demonstrated important roles in the prevention of various diseases. In the present study, the effect of a supplementation with cheese enriched with ω-3 fatty acids and CLA on the metabolism and lipid profiles of C57bl/6 mice was evaluated. In particular, the analyses were conducted on different tissues, such as liver, muscle, adipose tissue and brain, known for their susceptibility to the effects of dietary fats. Supplementing cheese enriched in CLA and ω-3 fats reduced the level of saturated fat and increased the content of CLA and ALA in all tissues considered, except for the brain. Furthermore, the consumption of this cheese resulted in a tissue-specific response in the expression levels of genes involved in lipid and mitochondrial metabolism. As regards genes involved in the inflammatory response, the consumption of enriched cheese resulted in a reduction in the expression of inflammatory genes in all tissues analyzed. Considering the effects that chronic inflammation associated with a high-calorie and high-fat diet (meta-inflammation) or aging (inflammaging) has on the onset of chronic degenerative diseases, these data could be of great interest as they indicate the feasibility of modulating inflammation (thus avoiding/delaying these pathologies) with a nutritional and non-pharmacological intervention.

## 1. Introduction

Human studies have shown that high-fat diets (HFDs) (>30% of energy from fat) can easily induce overweight/obesity, which is considered one of the most risk factors for chronic diseases, such as coronary heart disease (CHD), hypertension, type 2 diabetes (T2D) and cancers [1]. These effects are mainly related to the high level of saturated fatty acids (SFA), which are more abundant in animal products, such as milk and dairy products. However, these foods are rich in vaccenic acid (VA), conjugated linoleic acid (CLA) and polyunsaturated fatty acid (PUFA) ω-3, which have shown positive effects on human health [2]. Some of these fatty acids (FAs), such as CLA and VA, are synthesized and naturally accumulated in the milk and dairy products of ruminants, making these foods the main CLA and VA sources for the consumer. As demonstrated in several studies, the composition of dairy fat of ruminants is strongly affected by their diet [3,4]. Linoleic acid (LA) and α-linolenic acid (ALA) are the main dietary FAs modified by ruminal bacteria through the biohydrogenation process. Intermediate products of this process, such as VA and CLA, can bypass the rumen and accumulate in ruminant tissues and milk [5]. Moreover, LA and ALA, unmodified in the rumen, are also deposited in tissues and milk [5]. So, dairy products can be enriched with these FAs, in particular, VA, CLA and ALA, acquiring important nutraceutical properties for human health [4]. In particular, Mele and co-workers [4] obtained a CLA (+197%) and ALA (+250%) enrichment of Pecorino cheese (typical Italian sheep’s cheese) by supplementation of a ewe’s diet with extruded linseed (a natural source of ALA).

Cheeses enriched with CLA and ALA have shown positive effects on human health when incorporated into HFDs [3], for example, reducing plasma levels of endocannabinoid anandamide and LDL-cholesterol [3]. However, at present, little is known about the effect of enriched cheese on the regulation of lipid metabolism, particularly in certain tissues, such as skeletal muscle, adipose tissue, liver and brain.

Several studies have demonstrated that the positive effects of CLA and ALA are particularly evident in aging [6]. Aging is the main risk factor for all the chronic-degenerative diseases, such as CHD, cancer, T2D, arthritis, sarcopenia and neurodegenerative diseases, as well as for conditions such as frailty, which are the main causes of mortality in the western world. The impact of aging in the development of age-related diseases is partly caused by the changes in body composition that occur as adaptation to long lasting stresses. In particular, these changes include a decrease in lean mass and a progressive increase in fat mass, with modifications in its distribution, which negatively affect health status [7]. As an example, in the elderly, tissues such as heart, liver and skeletal muscle are characterized by increased deposition of ectopic fat, leading to an increased risk of insulin-resistance and cardiovascular diseases [8,9]. Furthermore, an increase in fat content in the bone marrow leads, together with the decrease in bone mineral density, to an increased risk of fractures [10]. Indeed, excessive lipid accumulation causes lipotoxicity in several non-adipose tissues [11] due an increase in the production of reactive oxygen species (ROS) and consequently of oxidative stress, leading to dysregulation in glucose and lipid metabolism [12]. Skeletal muscle, adipose tissue and liver are important organs for maintaining the homeostasis of energy metabolism. Skeletal muscle is the primary site for glucose uptake and storage and the reservoir of amino acids stored in the form of proteins, influencing energy and protein metabolism throughout the body [13]. White adipose tissue, which represents over 95% of the fat mass, is the main organ responsible for the synthesis and storage of lipids. The liver is responsible for the storage and synthesis of glucose and represents the hub of FA synthesis and lipid circulation [14]. The brain is also involved in the regulation of lipid metabolism; in fact, within the brain, lipid metabolism is tightly regulated to maintain neuronal structure and function and can signal the state of nutrients to modulate metabolism in key peripheral organs, such as the liver [15].

In the present study, we aimed to evaluate the nutritional effect of a supplementation with cheese enriched with CLA and ALA in C57bl/6 mice. In particular, we evaluated the effect of such supplementation on fat profiles and the expression levels of genes involved in lipid and mitochondrial metabolism (lipogenesis, lipolytic process, lipid drop development, Kennedy pathway, mitochondrial respiratory chain), as well as inflammation, in different tissues, such as the brain, liver, skeletal muscle and adipose tissue. The positive effects of PUFA ω-3 and CLA are undoubtedly known, as demonstrated by the extensive literature. However, little is still known about the effects of these fatty acids on various tissues, particularly in a high-fat cheese diet.

## 2. Materials and Methods

### 2.1. Experimental Design

Twenty-one male C57bl/6 mice six months of age with an initial weight of 27.0 ± 0.72 g were randomly assigned to three groups (seven animals per group) and fed for eight weeks with different diets: (i) control diet (CON), based on the Teklan global 14% protein maintenance diet (ENVIGO, Indianapolis, IN, USA) formulation containing 4% total fat, which provided 13% of the total energy; (ii) control diet + “Pecorino” cheese (CHE) (40%:60%); (iii) control diet + ALA and CLA-enriched “Pecorino” cheese (40%:60%) (ENR). Diets CHE and ENR were characterized by a high fat level, which provide 70% of total energy. The three diets were chosen for the following reasons: CON represents the diet usually used for mice, CHE constitutes the most appropriate control, while ENR represents the effective CLA- and ALA-enriched diet. The chemical composition of the three diets is presented in Table 1. Protein, lipid, carbohydrates and ash contents were determined with official AOAC (Association of Official Agricultural Chemists) methods, while the fatty acid profile was determined by gas-chromatographic analysis, as described later. Diets CHE and ENR were characterized by a high content of lipids and a lower level of carbohydrates. Moreover, CHE and ENR diets showed a higher amount of C16:0, C18:0, C18:1c9 and CLA (C18:2c9t11). The comparison between the two diets with cheese shows that the ENR diet is very rich in CLA and ALA, the levels of which exceed the thresholds (0.3 and 0.7%, respectively) of consumer well-being, as indicated by Mele et al. [4]. The CHE diet, on the other hand, is unable to guarantee these CLA and ALA levels, therefore it does not express the nutraceutical effects of these fatty acids. The dietary approaches proposed in this study were preferred to a simple addition of ω-3 mixed in the standard pellet diet as the added cheese is a dairy product already marketed and this approach should have allowed the mimicry, at least in part, of the consumption of cheese within a diet as varied as that of the consumer. The Pecorino cheese used in the CHE diet was produced according to the specification provided by the PDO consortium of “Pecorino Toscano”, while the cheese used in the ENR diet was obtained as described by Mele et al. [4]. Animals were housed in cages with food and water ad libitum, and they were exposed to a 12 h dark/light cycle. During the experimental period, the body weight of mice was recorded weekly.

Before sacrifice, mice were fasted for 12 h and their blood was collected, through the caudal vein, to perform a blood glucose test (Glucocard G meter, Menarini Diagnostics, Florence, Italy). Mice were euthanized by urethane injection (100 μg/kg of body weight) and cervical dislocation. Immediately after death, skeletal muscle (SM), brain (B), liver (L) and visceral adipose tissue (VAT) were taken and stored at −80 °C. After the mice died, intracardiac blood was collected in a blood collection tube with EDTA (BD Vacutaine). The blood was used for the measurement of the lipid panel (cholesterol, HDL, LDL and triglycerides) using the Cobas b101 instrument (Roche Diagnostic, Milan, Italy).

All experiments were performed according to the guidelines and protocols approved by the European Union (EU Council 750/2013; D.L. 26/2014) and by the Animal Research Ethics Committee of the University of Pisa, Italy. The authorization number from the Italian Ethical Committee was 12/2019-PR.

### 2.2. Lipid Analysis

Total lipids (TL) of whole samples (52 mg SM, 200 mg L, 150 mg B and 300 mg VAT) were extracted, according to the method described by Rodriguez-Estrada et al. [16], with some modifications. Briefly, samples were resuspended with a chloroform/methanol solution (2:1, *v*/*v*) and homogenized by ULTRATURRAX (IKA^®^ Werke GmbH & Co. KG, Staufen, Germany) for 60 s. After 2 h of incubation at 40 °C, samples were filtered to remove solid phase and then 2 mL of KCl 1M aqueous solution was added. The lower phase (apolar phase) was recovered and placed in a fresh tube. Lipid fractions were separated by thin layer chromatography (TLC) using a Silica Gel 60F254 10 × 20 cm with a mix of hexane and diethyl ether (70:30, v:v) as mobile phase solution. The spots corresponding to the phospholipids (PLs), triglycerides (TGs), free fatty acids (FFAs) and free and esterified cholesterol (FC and EC, respectively) were identified by comparing them with a mix of commercial standards (Sigma-Aldrich: code 17,810 for TGs, 690050C for PLs, MAK044 for FFAs, C4951 for FC and C9253 for EC). Each spot was scraped and recovered separately in a fresh tube with 1 mL of diethyl ether.

For TL and TG, FA methyl esters (FAMEs) were prepared using a methanolic NaOH basic solution (10%), according to Christie [17], with some modifications. Briefly, 1 mg of C19:0 as an internal standard was added to 20 mg of lipids, 1 mL of methanolic NaOH solution (10%) and 1 mL of hexane. Samples were incubated at RT for 1 min and subsequently FAMEs were extracted with 1 mL of hexane.

FAMEs of PLs and FFAs were prepared using a methanolic hydrochloric acid solution (10%), according to Christie [17], with some modifications. Briefly, 1 mg of C19:0 as an internal standard was added to 20 mg of lipids and 2 mL of methanolic hydrochloric acid solution (10%). Samples were incubated at 50 °C and subsequently FAMEs were extracted with 2 mL of hexane and 2 mL of water. After centrifugation at 3000× *g* for 5 min, the upper phase was transferred into a fresh tube, while the lower phase was washed twice with 2 mL of hexane. FAMEs were separated and identified using a GC-FID (GC 2000 plus, Shimadzu, Columbia, MD, United States), according to Mele et al. [4].

Finally, cholesterol fractions (EC and FC) were saponified, according to Sander et al. [18], and analyzed using a GC-FID (GC 2000 plus, Shimadzu, Columbia, MD, United States) equipped with a VF 1 ms apolar capillary column (30 m × 0.25 mm i.d., 0.25 μm film thickness; Varian, Palo Alto, CA, United States), with the carrier gas (hydrogen) flux at 1 mL/min, and the split ratio was 1:10, according to Serra et al. [19].

### 2.3. Gene Expression Analysis

Total RNA was extracted using different kits depending on the tissue: mirVanaTM miRNA Isolation Kit (Invitrogen, Thermo Fisher Scientific, Waltham, MA, USA) for SM, and RNeasy Lipid Tissue (QUIAGEN, Hilden, Germany) for B, L and VAT, according to the manufacturer’s instructions. RNA quantification was carried out with a NanoDrop 1000 Spectrophotometer (THERMO^®^Scientific). RNA was treated with DNase by TURBO DNA-freeTM kit (Ambion, Austin, TX, USA), following the manufacturer’s instructions, to completely remove genomic DNA contamination. Finally, cDNA was synthesized using iScriptTM cDNA Synthesis Kit (Bio-Rad, Hongkong, China), according to the manufacturer’s instructions. Real-time PCR was performed using iTaqTM Universal SYBR Green Supermix (Bio-Rad) and Rotor gene Q 6000 system (QIAGEN GmbH, Hilden, Germany). All data were normalized to GAPDH expression. All oligonucleotide pre-designed primers were from Bio-Rad. All information on these primers is available at the website www.bio-rad.com/PrimePCR (1 June 2017).

For this study, we considered the expression of the key genes involved in lipid metabolism, inflammation and the cellular cycle and mitochondrial metabolism (Appendix A).

### 2.4. Statistical Analysis

Multiple unpaired comparison tests were performed with one-way ANOVA followed by a Tukey’s post hoc multiple comparison test in order to check the effect of specific dietary lipids on lipid metabolism. Data were analyzed by the following linear model (SAS Institute Inc., Cary, NC, USA, 2010):y_ij_ = μ + T_i_ + A_j_ + ε_ij_(1)
where y_ij_ is the observed trait (fatty acids and cholesterol); μ is the overall mean; T_i_ is the fixed effect of the i^th^ treatment (CON; CHE; ENR); A_j_ = is the random effect of the j^th^ animal (1–21); and ε_ij_ is the random residual term.

Data are expressed as means ± SEM. Differences were declared significantly different at a *p*-value < 0.05.

## 3. Results

### 3.1. Effect of ENR Cheese Supplementation on Lipid Fraction Composition

The effect of the ENR cheese supplementation was evaluated in comparison with a standard chow diet (CON) and a diet supplemented with normal cheese (CHE). The mice in the three groups showed a similar initial weight and daily food intake (3.65 ± 0.35, 3.53 ± 0.43, 3.71 ± 0.43 g for CON, CHE and ENR, respectively), however at the end of the treatment, the body weight of ENR group mice was significantly higher with respect to other two groups (Table 2). Total cholesterol was significantly increased in CHE, while no differences were observed between CON and ENR. This result was due to a higher level of HDL-cholesterol in both cheese diets. LDL-cholesterol was decreased in both experimental groups with respect to CON. Both cheese supplementations are able to markedly reduce cardiovascular risk. On the contrary, TG levels in the blood did not change after cheese supplementation.

#### 3.1.1. Liver

The amount of fat was about 8.5%, which was mainly characterized by TGs (52%) and FFAs (30%), while PLs were the least represented component (8%) (Table 3). Cholesterol occurred principally as FC, while the EC was about 25 times lower (Table 3).

Total lipids, TGs and PLs did not show significant differences between the three groups, while FC decreased by 43% in CHE and ENR groups. Esterified cholesterol, on the other hand, increased significantly (more than five times) in the ENR mice only.

Regarding the effect of diet on total FA profile, we found a significant increase of SFAs (+47%) and MUFAs (+31%) in CHE and ENR group mice. Conversely, we observed a significant reduction of PUFAs (−46%) in the same groups (Table 2). The higher level of SFAs was mainly due to the increase in C16:0 (+21% and 13% for mice in the CHE and ENR groups, respectively) (Appendix A). There was also a fairly significant increase of C18:0 (+100%) in mice in both experimental groups. Among the other SFAs, there was a significant increase of medium and short chain fatty acids in the CHE and ENR groups (Appendix A). As for the PUFAs, the main changes occurred for linoleic acid, which significantly decreased by 59% and 61% for CHE and ENR, respectively. Importantly, C18:2c9t11 (CLA) increased dramatically (+94-fold) in mice of the ENR group (Table 3). On the contrary, we observed a significant reduction of C20:5n3 (−59% and −55% for CHE and ENR, respectively) and C22:3n3 (−78% and −100% for CHE and ENR, respectively) (Appendix A).

As for MUFAs, the supplementation with cheese resulted in an increase of oleic acid (+50%), of C14:1c9 of (+487%) and of C20:1c11 (+320% for the CHE group and +388% for the ENR). It is interesting to note that, in both experimental groups, all the products of the rumen biohydrogenation process increased: C18:1t6-8 (+10 times), C18:1t9 (+30 times), C18:1t10 (+30 times), C18:1t11 (+50 times) and C18:1c11 (+57% for CHE and +41% for ENR) (Appendix A).

#### 3.1.2. Brain

The amount of fat was about 10%, which was mainly characterized by PLs (65%) and TGs (22%), while FFAs were the least represented component (8%) (Table 4). Cholesterol occurred principally as FC, while EC was not revealed (Table 4).

All lipid fractions did not show significant differences between the three groups. Also in this case, regarding the fatty acid profile, no effects due to a high-energy diet with cheese supplementation were observed.

#### 3.1.3. Skeletal Muscle

The amount of fat was about 21%, which was mainly characterized by TGs (85%) and PLs (14%), while FFAs were the least represented component (3%) (Table 5). Cholesterol occurred principally as FC, while EC was about seven times lower (Table 5).

The total lipids fraction was not affected by the two experimental diets (Table 4). In contrast, mice of the ENR group showed a significant increase in PLs (+102%) and a reduction in TGs (−12%). In both experimental groups, cheese consumption resulted in a significant increase in FFA (+11 times). The two experimental groups also displayed a reduction in FC content (−43%), while no significant differences were found for esterified cholesterol.

The supplementation with cheese showed a significant increase in SFAs, in particular, CHE (+54% and +34% in CHE and ENR, respectively); in contrast, MUFAs decreased by 22% and 31% in CHE and ENR, respectively (Table 5). The level of PUFAs significantly decreased by 42% in CHE, while no differences were observed in ENR. Notably, PUFAn-6 decreased by 48% and 37% in CHE and ENR, respectively. PUFAn-3, on the other hand, increased principally in ENR (+233%); however, we also observed a lower but significant increase (+23%) in CHE (Table 5).

The higher level of SFAs in CHE and ENR was mainly due to the increase of short and medium chain FAs deriving from dairy product consumption, as also observed in the liver (Appendix A). The most diet-influenced MUFAs were principally C18:1 trans isomers (C18:1t9, C18:1t10 and C18:1t11) and oleic acid (Appendix A). Finally, many PUFAs showed a significant change with diets containing cheese. In particular, we observed a reduction of C18:2n-6 (−68%), and an increase of n-6 and n-3 PUFAs, deriving from LA and ALA elongation (Appendix A).

#### 3.1.4. Adipose Tissue

The amount of fat in the adipose tissue was about 58%, which was mainly characterized by TGs (95%), while PLs represented only 1% (Table 6). Cholesterol occurred principally as FC (7 mg/100 g of total lipids). Lipid fractions were not affected by the two experimental diets (Table 6).

The supplementation with cheese showed a significant increase in SFAs (+41% for both experimental groups) and MUFAs (+15% and +31% in CHE and ENR, respectively). On the contrary, PUFAs did not showed differences (Table 6).

The higher level of SFAs in CHE and ENR was mainly due to the increase of the following fatty acids: C8:0 (+431%); C10:0 (+402%); C12:0 (+53-fold); C14:0 (+421%); C15:0iso (+175% only in ENR); C15:0ante (+450%); C15:0 (+366%); C16:0iso (+97%); C17:0iso (+87%); C17:0 (+153%); (C18:0 +54%); C21:0 (+241%, only in ENR) (Appendix A). The MUFAs affected by the high-fat diet with cheese supplementation were: C14:1c9 (+534%); C16:1c9 (+78%); C17:1c9 (+55%); C18:1t6-8 (+216% and +409% in CHE and ENR, respectively); C18:1t9 (+634% and 1213% in CHE and ENR, respectively); C18:1t10 (+496% and +1590% in CHE and ENR, respectively); C18:1t11 (+63-fold and +121-fold in CHE and ENR, respectively); C18:1c9 (+18%, only in ENR); C18:1c11 (+489%, only in ENR); C18:1c12 (+25-fold and +98-fold in CHE and ENR, respectively). Finally, many PUFAs showed a significant change with diets containing cheese: C18:2n-6 (−68%); C18:3n-6 (+326% in ENR); C18:2c9t11 (+33-fold and +94-fold in CHE and ENR, respectively) (Appendix A).

### 3.2. Effect of Diet on Gene Expression

The effect of diet on the expression of some genes involved in lipid and mitochondrial metabolism and in the control of stress and inflammation is reported in Table 7. The potential pathway of these genes is summarized in Figure 1.

#### 3.2.1. Liver

We observed a significant change in the expression of all perilipin genes (*PLIN*) in the groups supplemented with cheese, compared to the control group: *PLIN2* showed a significant increase (+68%) in ENR and a reduction (−23%) in CHE; *PLIN3* expression increased (+12%) in CHE and decreased (−42%) in ENR; *PLIN4* is the gene that showed the greatest variability of expression, with a significant increase in the CHE (+1037%) and ENR groups (+1620%); *PLIN5* increased (+ 38%) in the ENR group and decreased (−83%) in the CHE group.

All lipogenic genes were significantly under-expressed with both supplementations: *ACACA* −46% for both cheese-fed groups; *ACACB* −78% in CHE and −68% in ENR; *FASN* −78% in CHE and −91% in ENR; *SCD* −66% in CHE and −40% in ENR; *ACLY* −64% in CHE and −85% in ENR. A particular effect was observed for the *SREBP1* gene, which decreased in the mice of the ENR group (−94%) but increased in the CHE group (+ 338%).

Genes involved in the ceramide biosynthesis pathway, such as *CERK*, *CERS6* and the Kennedy pathway, such as *ETNK1* and *CEPT-1*, were not influenced by the cheese diet; significant changes were in fact detected.

Three genes involved in the regulation of inflammatory processes showed a significantly lower expression in groups fed with cheese: *PRKCQ* −13% in CHE and −46% in ENR; *TRP53* −54% in CHE and −78% in ENR; *FGF21* −70% in CHE and −52% in ENR.

The effect of cheese was also observed in mitochondrial metabolism, where a significantly higher expression was observed for the following genes: *PGC1A* +57% in the CHE group and +82% in the ENR group; *OPA-1* +57% in CHE and +106% in ENR. On the contrary, expression of the following genes was lower: *IL1B* −65% for both groups; *COX10* −24% in the CHE group and −41% in ENR; *UCP2* −56% in CHE and −33% in ENR; *ATP5G1* −15% in CHE and −42% in ENR; *MFN1* −51% in CHE and −57% in ENR.

#### 3.2.2. Brain

Among the genes encoding for *PLINs*, only *PLIN2* and *PLIN3* showed lower expression in the ENR group, while CHE showed a level of expression similar to CON.

We also observed a different expression of some genes involved in lipid synthesis. *FASN* expression (+129%) was significantly higher in ENR, while no effect was observed in the CHE group; *SCD* and *ELOVL2*, on the other hand, were not as highly expressed (−10% and −79%, respectively) in mice of the ENR group. The expression of *ACLY* and *SREBP1* decreased in the two experimental groups with respect to the control. Three genes involved in glycerolipid metabolism decreased significantly in the experimental groups: *DGAT1* (−37% and −79% in CHE and ENR, respectively); *DGAT2* (−47% and −72% in CHE and ENR, respectively); *LPIN1* (−15% and −62% in CHE and ENR, respectively).

A significant effect was observed for the following genes involved in the regulation of inflammatory processes: *PRKCQ* (−21% and −77% in CHE and ENR, respectively) and *TRP53* (−47% and −82% in CHE and ENR, respectively).

Genes involved in the Kennedy pathway, showing a different expression were *ETNK1* (−62% in both experimental groups), *PCYT2* (−39% and −70% in CHE and ENR, respectively) and *CERK* (−20% and −51% in CHE and ENR, respectively). Finally, *UCP2* and *UQCR10* genes, involved in mitochondrial metabolism, showed a significant different expression (−66% and −49%, respectively, in both experimental groups).

#### 3.2.3. Skeletal Muscle

Among the *PLINs*, *PLIN2* showed a tendency to decrease with cheese consumption, although this effect was not statistically significant. On the contrary, the other three *PLINs* showed significant changes: *PLIN3* (−51% and −18% in CHE and ENR, respectively); *PLIN4* (−62% for both experimental groups); *PLIN5* (−73% for both experimental groups). Among the other genes involved in lipid synthesis, those that showed a different expression were: *SCD* (−75%), *DGAT2* (−57%), *ACACA* (−62% in the ENR group) and *LPIN1* (−42% in the CHE group).

Regarding the genes of the Kennedy pathway, the gene *ETNK1* showed reduced expression (−60%) in both groups. On the contrary, the *PCYT2* gene was more highly expressed (+81%) in ENR. *PISD* expression only decreased in ENR. *CERK* also decreased its activity in the groups with cheese supplementation (−50% for CHE and −68% for ENR), while *CEPT* showed a significant decrease (−62%) in ENR.

The mitochondrial metabolism genes that showed significant differences were: *OPA-1* (−25% for the ENR group); *PGC1A* (+100% and +43% in CHE and ENR, respectively); *UCP2* (−88%) and *UCP3* (−77%); *UCP3*; *MFN1* (+ 565% in CHE).

#### 3.2.4. Adipose Tissue

We observed a significant change in the expression of *PLIN4*, which showed a significant decrease (−81%) in the CHE and ENR groups.

Lipogenic genes that showed significant differences among the three groups were: *SREBP1* (+338% only in the CHE group), *ACACB* (−72% in both experimental groups), *FASN* (+4-fold and +11-fold in the CHE and ENR groups, respectively), *GPAM* (−84% in both experimental groups), *DGAT2* (+66% in the CHE group), *LPIN1* (−74% in both experimental groups), *ACLY* (+123% in the CHE group), *CERK* (−41% and −67% in the CHE and ENR groups, respectively), *PCK1* (−82% in both experimental groups), *PRKAA1* (−63% in both experimental groups), *ATGL* (−85% in both experimental groups), *CPT1b* (not expressed in the ENR group).

Three genes involved in the regulation of inflammatory processes showed a significant differential expression in the CHE and ENR groups: *TRP53* (−73% in the ENR group) and −78%, *FGF21* (−88% in both experimental groups) and p21 (−81% in both experimental groups).

The effect of cheese in a high-energy diet was also observed in mitochondrial metabolism, where significantly lower expression was observed for the following genes: *UCP2* (−67% in both experimental groups), *NDUFS3* (−77% in the ENR group), *UQCR10* (−38% and −95% in CHE and ENR, respectively), *COX10* (−53% in the ENR group), and *KLOTOB* (−89% in both experimental groups).

## 4. Discussion

Three types of diets were compared: one was the standard pellet diet for mice (CON); the second consisted of CON with pecorino cheese (CHE), which constituted the most appropriate control; finally, the third consisted of the CON diet with pecorino cheese enriched with ω-3 and CLA (ENR). This approach was preferred to a simple addition of ω-3 mixed in the pellet as the added cheese is a dairy product already marketed and this approach should have allowed the mimicry, at least in part, of the consumption of cheese within a standard varied diet. Furthermore, the two diets supplemented with cheese were characterized by high levels of energy intake, so they could be considered as HFDs. This experimental model has therefore allowed us to verify whether supplementation with ω-3 PUFA and CLA is able to mitigate or suppress the effects of HFDs, which consists of an excessive caloric intake and a high amount of SFAs [20]. A first interesting result was the improvement of the blood lipid panel in the CHE and ENR diets, with a significant reduction in LDL cholesterol levels and an increase in HDL levels, which contributes to a reduction of cardiovascular risk in humans, as previously demonstrated by Pintus et al. [3]. The data show how the inclusion of cheese in a diet rich in fat allows the counteraction of the negative effects of a HFD, which are often associated with the development of metabolic disorders.

### 4.1. Lipid Profile

The lack of changes in the brain is justified by the composition of its lipid profile. It is known that the brain is an organ rich in PLs, which play an important role in regulating membrane fluidity [21]. The alteration of the PL profile (in particular, for the level of unsaturation) can affect this aspect by modifying the interaction with other contiguous cells, as well as enzymes and membrane receptor activity [22]. Therefore, there could be a metabolic regulation system, not yet demonstrated, which may counteract PL homeostasis due to external factors, such as diet. However, further investigations are necessary to deepen this regulation system. This would justify the lack of an effect of the diet on the normal lipid profile of the brain, while in the liver a tendency to reduce the level of TGs was observed in mice belonging to the ENR group, although the results were not significant. This trend agrees with Backes et al. [23], whose report stated that a diet enriched with ω-3 tends to decrease TGs in the liver. The lack of significance may be due to the particularities of hepatic metabolism, where there is a continuous flow of both endogenous and exogenous lipids, which can partly mask the effect reported by Backes et al. [23]; on the contrary, this reduction was seen in the muscle.

In skeletal muscle, we observed a significant reduction of TGs in the ENR group. This is linked to a reduction in lipogenesis activity and an increase in β-oxidation activity, as also noted by Backes et al. [23]. In particular, consumption of the ENR diet led to a significant increase in FFAs in the muscle, associated with a possible increase in lipolytic activity which increases the β-oxidation process in accordance with the results of Ibeagha-Awemu et al. [24] and Yan et al. [25]. Furthermore, we observed a significant decrease in FC in both liver and muscle and a simultaneous increase in EC in the liver of mice fed the ENR diet. The increase in hepatic EC may be associated, although this is not proven by experimental data, with an increase in the activity of the ACAT1 protein. This increase is associated with some positive effects for human health, first of all, the increase in EC, which is referred to a lower availability of FC for the production of LDL cholesterol in plasma [26]. Furthermore, a high level of cholesterol in the liver is also associated with a reduction in liver fibrosis, as demonstrated by Tomita et al. [27].

In all organs, we observed an increase in levels of SCFAs and MCFAs in the cheese-fed mice. These FAs are an indicator of the consumption of dairy products, particularly those resulting from the secretion of the mammary gland of ruminants (C4-C16). These FAs are synthesized exclusively in the mammary gland of ruminants thanks to a particular role of the mammary *FASN* gene, which is able to stop the synthesis of FAs completely at C16:0 and for which it can accumulate C8:0, C10:0 and C14:0 [2]. Other FAs that are significantly increased in the four organs with cheese consumption are all trans C18:1 isomers. These FAs are intermediate or final products of the rumen biohydrogenation process, and monogastric animals can only obtain them by consuming foods derived from ruminants [4]. The increased accumulation of these isomers in mice of the ENR group is due to the higher level of α-linolenic acid, which is one of the substrates of biohydrogenation. High levels of biohydrogenation led to the accumulation of VA and consequently to a proportional increase of CLA. This aspect is interesting because CLA is not present in CON diets, demonstrating the fact that if products of animal origin are not consumed, this important FA is not present along with all the beneficial activities that it permits. The mice of the ENR group ate, on average, about 0.4 g of CLA per day, an amount close to the values obtained to observe effective health effects. In fact, it has been shown that hamsters who ate 0.5 g of CLA per day for 12 weeks showed a reduction in aortic problems [28] and glucose tolerance [29].

An FA that showed a higher level in the ENR group was gamma-linolenic acid (GLA, C18:3n-6). This FA is an ω-6 found in human milk and several oilseeds [30]. Clinical studies have shown that its presence in the diet leads to an alteration of different inflammatory responses [30]. The increase in this fatty acid was observed only in the muscle (both in the TGs and in the PLs).

### 4.2. Gene Expression

Skeletal muscle, liver, brain and adipose tissue are the main site of whole-body FA and carbohydrate oxidation [31]. The relative contribution of these fuels to the energetic demands of the tissues is subject to complex regulation at multiple levels, including substrate availability, hormonal concentrations and the allosteric regulation of enzyme activities by intracellular metabolic intermediates [32]. In normal physiological conditions, glucose availability and flux exert the dominant effect on the oxidized fuel mix [33]. However, an HFD elevates the contribution of FAs to oxidative metabolism, especially in obese subjects [34]. In addition to their role as important oxidative substrates, dietary FAs regulate the expression of many genes [35]. Moreover, the response of those genes is central to the regulation of fatty acid transport and mitochondrial β-oxidation, whose actions are likely to be pivotal in the increased capacity to oxidize fatty acids. The consequence of oxidative stress is converging in the induction and regulation of inflammation, as described by Vitale et al. [36]. For this reason, we focused the analyses on particular genes, as reported in Appendix A, concerning lipid metabolism, the mitochondrial oxidation system and the inflammation pathway.

#### 4.2.1. Skeletal Muscle

An important aspect of SM lipid metabolism is the profile of PLINs, which represent the main proteins involved in the regulation of lipid droplet (LD) homeostasis as well as lipolysis and lipogenesis [37]. Recent findings show that modifications of PLIN expression dysregulate intracellular lipid deposition, lead to the accumulation of diacylglycerols or ceramides, impair cellular function and cause lipotoxicity [37]. Our data suggest that both the ENR and CHE diets do not affect the expression of *PLIN2*, which is considered the marker of LD and thus of the accumulation of fat in non-adipose tissues. Conversely, we revealed an effect on the expression of other proteins belonging to the PLIN family, such as *PLIN3* and *PLIN5*. It is well known that *PLIN3* and *PLIN5* are involved in lipid oxidation and play a role in the regulation of lipolysis in skeletal muscle [38,39]. In particular, *PLIN5* is a fundamental protein for the mobilization of FA, as it acts as a carrier for LD towards the mitochondria, where FAs undergo β-oxidation [38]. Although knowledge about *PLIN3* in skeletal muscle is poorer with respect to *PLIN5*, recent studies suggest that *PLIN3* plays a key role in regulating lipolysis by interacting with lipases, similarly to that found for *PLIN5* [39]. *PLIN5* expression decreased with both experimental diets, suggesting that an increase in dietary fat inhibits FA mobilization. The expression of *PLIN3* is decreased only in CHE but not in ENR. It can therefore be hypothesized that supplementation with PUFA ω-3 and CLA induces the mobilization of FAs and therefore β-oxidation by the activity of *PLIN3* to compensate for the failure of *PLIN5*.

An interesting effect was observed for *CEPT-1* expression, which is involved in the Kennedy pathway for the biosynthesis of phosphatidylcholine and phosphatidylethalonamine [40]. The ENR diet reduced the expression of *CEPT-1*, and this probably can lead to an improvement in insulin sensitivity. Indeed, it has been observed that obese mice fed an HFD show reduced insulin sensitivity and increased expression of the *CEPT-1* gene [40].

*DGAT1* and *DGAT2* are responsible for the synthesis and storage of TGs, and levels of expression decreased with the ENR diet. *DGAT1* knockout mice have been shown to develop resistance to diet-induced obesity and have improved insulin sensitivity due to reductions in the expression levels of genes involved in lipid uptake and oxidation, thus preventing lipotoxicity [41]. The limited accumulation of triglycerides has also been shown to protect against fatty acid-induced lipotoxicity [42]. Moreover, we observed a reduced expression of *ACACA* and *SCD* genes in the ENR diet, suggesting reduced liponeogenesis. This is likely due to the introduction of essential FAs through the diet which are stored in the muscle instead of activating de novo synthesis. The inhibitory effect of PUFAs on liponeogenesis genes has been demonstrated in both mice and humans [43].

As regards the genes of mitochondrial metabolism, interesting data were observed for the *UCP2* and *UCP3* genes linked to the decoupling of substrate oxidation from ATP synthesis. Both genes showed increased expression in the ENR group compared to CHE. Decreased mitochondrial decoupling in SM has been shown to lead to increased oxidative stress and to the development and progression of certain diseases, such as T2D [44,45]; therefore, PUFA ω-3 could limit this damage by increasing the expression of genes for Ucp.

Both CHE and ENR groups reduced the expression levels of ceramide kinase (CERK) suggesting a reduction in inflammation levels [46]. CERK is involved in the regulation of ceramide phosphorylation and in the production of ceramide-1-phosphate (C1P), a sphingolipid which is commonly implicated in inflammation. A genetic ablation of CERK in mouse embryonic fibroblasts reduces the production of eicosanoids, well known for their involvement in pro-inflammatory processes [47]. This result is also confirmed by the lower expression of *NFATC2* and *PRKCQ* genes in the ENR group, two genes closely related to inflammation [48]. A similar result was also obtained for the *TRP53* gene, which showed lower expression in the ENR than in the CHE group. TRP53 is best known as the guardian of the genome, involved in DNA repair phenomena, apoptosis and, together with its p21 downstream, cell cycle blockade and cell senescence. Studies have shown that mice with high TRP53 activity are resistant to cancer but show signs of accelerated aging and die prematurely [49,50]. Regarding the correlation with inflammation, p53 and p21 are known to be potent inducers of cellular senescence, in turn a potent source of pro-inflammatory mediators (the so-called secretory phenotype associated with senescence, SASP). Furthermore, p53 regulates numerous processes related to energy and lipid metabolism, such as glycolysis, oxidative phosphorylation, lipolysis, lipogenesis, β-oxidation, gluconeogenesis and glycogen synthesis [51]. Its activation following obesity or a high-calorie diet would therefore contribute to the onset of inflammation [52]. The fact that the ENR diet induces a reduction of TRP53 expression compared to CHE suggests that ω-3 and CLA supplementation reduces the levels of inflammation induced by HFD.

#### 4.2.2. Liver

In the liver, as regards the expression of PLINs, most interesting was the very high expression level of *PLIN4* in both cheese-enriched diet groups, with a significantly higher expression level in ENR with respect to CHE. The role of *PLIN4* in the liver, as well as in other organs, is not still clarified. However, a study performed in hepatocyte-specific *PLIN2* knockout mice fed with a high-fat western diet (WTD), in agreement with our results, interestingly revealed increased protein expression levels of PLIN4 [53]. Moreover, another study in which C57BL/6 male mice were fed an HFD and simultaneously treated with matcha green tea for six weeks to prevent the excessive accumulation of hepatic lipids, showed higher levels of PLIN4 gene expression than mice fed only with an HFD, while mice fed with an HFD along with matcha green tea supplementation did not accumulate triglycerides in the liver concomitant to liver downregulation of PLIN4 expression levels [54]. Taken together, these results suggest that PLIN4 could play a critical role in liver lipid accumulation in the case of HFDs. However, at present, few data are available on PLIN4 in the liver and other experiments are necessary to understand the beneficial or detrimental role of PLIN4 in this organ.

Genes related to de novo synthesis of FAs, such as *ACACA*, *ACACB*, *FASN, SREBP-1* and *ACLY*, showed reduced expression in ENR. Decreased expression of *ACACA* or *ACACB* is associated with liver cancer suppression, as demonstrated in rats [55]. In contrast, Nelson et al. [56] observed a completely opposite effect for *ACACA* and *ACACB* in mice with a double knockout. Downregulation of the *FASN* gene is also associated with positive effects on human health. Indeed, the high level of this gene is associated with several types of cancer, as well as chemoresistance and metastasis [57]. The lower expression of *SREBP-1* in ENR was related to dietary PUFA ω-3, with a consequent reduction of liver damage, as demonstrated in a previous work [58]. Therefore, an ENR-associated decrease of SREBP-1 expression levels may be related to a beneficial effect of PUFA ω-3-enriched cheese in the liver. The higher level of *PCK1* expression in the ENR group compared to CHE increases the health properties of PUFA ω-3-enriched cheese in the HFD diet. This gene is involved in hepatic gluconeogenesis to ensure that glucose production rates match the body’s demands for glucose [59]. Recent studies have demonstrated a tumor suppressor role for *PCK1* in hepatocellular carcinoma [50]. In particular, several studies have shown that *PCK1* promotes cell death, especially in conditions of reduced glucose levels [60]. Moreover, a positive effect of PUFA ω-3-enriched cheese was the increase in the expression of PGC-1α, which was associated with a reduction in hepatic triacylglycerol levels [61].

Another interesting effect of the ENR diet was the lower expression of *GDF15*, which is the molecule that most correlates with age [62,63]. As is known, biological age is linked to several processes, such as mitochondrial dysfunction, oxidative stress, protein glycation, inflammation and hormonal changes. Many of these stresses are induced by increased expression of *TPR53* or *EGR-1**,* which in turn induce the expression of *GDF-15* gene [62]. High levels of *GDF-15* are associated with increased mortality linked to cardiovascular diseases, such as heart problems, coronary artery disease, atrial fibrillation, T2D and cognitive impairment [62,63]. Increased *GDF*-15 expression has been also associated with many cancers affecting the mammary gland, colon, pancreas and prostate. Therefore, the reduction in the expression of this gene in the diet with enriched PUFA ω-3 cheese could suggest a positive effect on the health of the consumer. The positive effect of enriched cheese on the liver, and likely skeletal muscle, is also indicated by the reduction in expression levels of *CERK, NFATC2* and *PRKCQ*, suggesting a positive role of PUFA ω-3 in counteracting inflammation. It is well known that a decrease of *NFATC2* expression is associated with a reduction of the expression of cytokines in T cells, in particular IL-2, IL-3, IL-4 and TNF-alpha; the latter, in particular, is a pro-inflammatory cytokine associated with obesity and insulin resistance [64]. Lack of expression of this gene in mice is associated with insulin resistance even with an HFD [65]. Our data suggest the positive role of enriched PUFA ω-3 and CLA cheese in HFDs as well. At the same time, we also observed a reduction in *PCYT2*, the key enzyme in the biosynthesis of phosphoethanolamine in the Kennedy pathway. The biosynthesis of membrane phospholipids, including phosphoethanolamine, is of fundamental importance for cell growth. High levels of phosphoethanolamine plasmalogens have been observed to be associated with an increased predisposition to ovarian cancer [66]. A similar result was obtained for the *TRP53* gene, whose expression levels were reduced more with the ENR diet compared to the levels shown with the CHE. The fact that the diet of the ENR group induces a reduction in p53 compared to the CHE group reinforces our hypothesis that ω-3 supplementation reduces the levels of inflammation induced by a high-fat diet.

#### 4.2.3. Brain

In the brain we observed a significant reduction of *TRP53*, *PRKCQ* and *CERK* levels in the ENR group. In particular, it has been shown that *CERK* is involved in neurodegenerative diseases, inducing toxicity in rat astrocytes and neurons [67].

Regarding lipid metabolism, we observed a significant decrease of *PLIN2* and *PLIN3* expression in the ENR group. Very recently, it was demonstrated that PLIN2 in the brain is the only PLIN family member that is modulated with age and neuroinflammation, suggesting that PLIN2 may be connected with the brain, aging and inflammation [68]. Therefore, the decrease in PLIN2 expression in the ENR group suggests that PUFA ω-3 can prevent lipid accumulation and, consequently, possibly lipotoxicity and thus inflammation. It has been shown that lipotoxicity can also occur in the central nervous system in some neurodegenerative disorders [69]. Similarly, mice fed with the ENR diet showed a decrease in the expression level of *SCD*. A previous study has shown that high levels of *SCD* are associated with Alzheimer’s disease, resulting in high levels of MUFAs [70]. The expression of the *ACLY* gene was also reduced in the ENR diet, with a healthy effect. Elevated levels of *ACLY* have been discovered in patients with brain tumors [71], so PUFA ω-3 could contribute to a reduction in the risk of developing these tumors. FASN, a key enzyme in de novo lipogenesis, is highly active in neural stem cells and its deletion impairs adult neurogenesis [72]. Its levels tended to increase in the diet of the ENR group compared to the CHE group, suggesting the positive effect of enriched cheese. As for mitochondrial metabolism genes, UCP2 showed lower levels of expression with the ENR diet. UCP2 deficiency has been shown to cause an increase in resistance to cerebral ischemia, with an increase in cerebral neuronal antioxidant status [73].

#### 4.2.4. Adipose Tissue

We observed a significant reduction in the expression of the *CERK* and *TRP53* genes in the ENR group, suggesting that PUFA ω-3 helps to counteract inflammation also in VAT. Unlike muscle, however, the expression of the *NFATC2* and *PRKCQ* genes did not change. *PLIN1* expression tended to decrease with the ENR diet, suggesting that PUFA ω-3 prevents lipid accumulation despite the HFD. *DGAT2* showed a reduction in expression in ENR compared to CHE. Inhibition of triglyceride synthesis can improve health status associated with obesity, since endogenous synthesis of triglycerides leads to an increase in the accumulation of lipids in adipose tissue [74]. Among the genes involved in lipogenesis, *ACACB* and *FASN* are the genes that have shown greater variability with HSD. *ACACB*, which regulates the oxidation of FAs, showed a lower expression in the ENR group. *ACACB* knockout mice have been shown to be protected from HFD-induced obesity and T2D through increased FA and glucose oxidation activity and increased lipolysis, thus promoting the maintenance of insulin sensitivity [75]. On the contrary, the expression of *FASN* only increased in ENR. This increase is probably due to the simultaneous increase of SREBP-1 expression, a transcription factor that regulates the synthesis of fatty acids [76]. Overexpression of SREBP-1 has been shown to activate genes linked to the FA biosynthesis pathway, leading to an increase in the accumulation of lipids in adipocytes, resulting in hyperglycemia and hypertrophy that causes an inflammatory reaction [77].

## 5. Conclusions

This study highlighted two important phenomena: (i) ω-3- and CLA-enriched cheese in an HFD reduces the expression of genes involved in inflammation and (ii) promotes metabolism involved in the prevention of cancer and cardiovascular disease. Moreover, our study shows how a diet supplemented with ω-3 fatty acids-enriched cheese determines changes in gene expression, suggesting an overall nutraceutical effect in all the tissues analyzed. Considering the effects of inflammation as a consequence of high-calorie and high-fat diets (metaflammation) or being associated with aging (inflammaging) on the onset of several chronic degenerative diseases, these data are potentially of great clinical interest, as they suggest the possibility of avoiding/delaying these pathologies (and aging itself) by means of a nutritional and non-pharmacological intervention. Further studies are needed to confirm these results, including longitudinal studies to assess the effects of such dietary supplementation on animal lifespans and disease onset.

## Figures and Tables

**Figure 1 foods-11-00398-f001:**
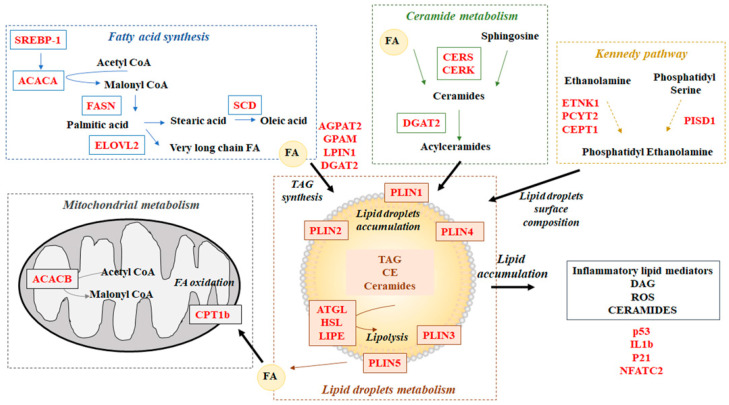
Metabolic pathway of genes considered in the study. FA = fatty acids; TAG = triacylglycerol; CE = Esterified Cholesterol; DAG = Diacylglycerol; ROS = Reactive Oxygen Species.

**Table 1 foods-11-00398-t001:** Composition of the three diets.

	CON	CHE	ENR
Protein (%)	14.30	21.81	19.95
Lipid (%)	4.00	23.07	21.89
Carbohydrates (%)	48.00	19.38	19.38
Ash (%)	4.70	3.54	2.77
Energy (cal/g)	278.23	377.62	359.02
C16:0	0.50	3.03	2.41
C18:0	0.10	1.49	1.49
C18:1c9	0.70	2.85	3.17
C18:1c9c12	2.00	1.10	1.11
C18:2c9t11	0.00	0.13	0.73
C18:1c9c12c15	0.10	0.12	0.33

CON = control diet; CHE = control diet + pecorino cheese; ENR = control diet + enriched pecorino cheese. The table shows the main fatty acids.

**Table 2 foods-11-00398-t002:** Effect of the diet on body weight and blood composition.

	CON	CHE	ENR	SEM	*p*-Value
Initial body weight (g)	26.00	25.00	28.00	1.50	0.450
Final body weight (g)	26.50 ^B^	26.00 ^B^	32.00 ^A^	1.45	0.009
Total Cholesterol (mg/dL)	78.00 ^B^	100.00 ^A^	78.00 ^A^	3.27	<0.001
Cholesterol HDL (mg/dL)	25.50 ^C^	72.00 ^A^	40.00 ^B^	2.50	<0.001
Cholesterol LDL (mg/dL)	25.00 ^A^	15.00 ^B^	12.50 ^B^	2.00	0.004
Cardiovascular risk (Total Chol./Chol. HDL)	2.80 ^A^	1.40 ^B^	1.50 ^B^	0.10	0.032

SEM = standard error of the mean. ^A,B,C^ Differences on the row *p* < 0.01.

**Table 3 foods-11-00398-t003:** Effect of the diet on the lipid profile of the liver.

	CON	CHE	ENR	SEM	*p*-Value
Total Lipids (g/100 g of liver)	9.47	8.20	8.26	0.98	0.762
Phospholipids (g/100 g TL)	11.12	5.59	8.40	2.75	0.237
Triglycerides (g/100 g TL)	50.47	55.56	49.99	4.15	0.321
Free Fatty Acids (g/100 g TL)	37.70	38.09	40.73	4.93	0.550
Free Cholesterol (mg/100 g TL)	193.01 ^a^	94.88 ^b^	109.88 ^b^	37.37	0.025
Esterified Cholesterol (mg/100 g TL)	4.38 ^b^	4.50 ^b^	28.28 ^a^	0.68	<0.001
Saturated Fatty Acids	23.19 ^B^	30.28 ^A^	30.38 ^A^	1.62	<0.001
Monounsaturated Fatty Acids	41.99 ^C^	48.26 ^B^	55.05 ^A^	1.75	<0.001
Polyunsaturated Fatty Acids	33.36 ^A^	13.14 ^B^	14.89 ^B^	1.20	<0.001
Polyunsaturated Fatty Acids n-6	31.79 ^A^	10.73 ^B^	10.02 ^B^	1.07	<0.001
Polyunsaturated Fatty Acids n-3	1.20	0.89	1.03	0.40	0.483
n-6/n-3	26.53 ^A^	12.12 ^B^	9.75 ^C^	0.33	0.005
Conjugated Linoleic Acid	0.04 ^C^	1.22 ^B^	3.40 ^A^	0.12	<0.001

SEM = standard error of the mean; TL = total lipids. ^a,b^ Differences on the row *p* < 0.05. ^A,B,C^ Differences on the row *p* < 0.01.

**Table 4 foods-11-00398-t004:** Effect of the diet on the lipid profile of brain.

	CON	CHE	ENR	SEM	*p*-Value
Total Lipids (g/100 g of brain)	8.88	10.65	11.33	1.28	0.156
Phospholipids (g/100 g TL)	63.27	62.87	70.14	3.11	0.347
Triglycerides (g/100 g TL)	25.37	25.95	22.64	3.23	0.537
Free Fatty Acids (g/100 g TL)	10.30	10.65	6.81	2.04	0.103
Free Cholesterol (mg/100 g TL)	897.78	720.15	1217.02	258.95	0.201
Esterified Cholesterol (mg/100 g TL)	-	-	-	-	NE
Saturated Fatty Acids	43.63	50.33	44.32	3.21	0.503
Monounsaturated Fatty Acids	28.47	30.39	27.76	2.06	0.303
Polyunsaturated Fatty Acids	17.96	20.39	20.19	1.96	0.203
Polyunsaturated Fatty Acids n-6	13.64	13.27	13.22	1.21	0.364
Polyunsaturated Fatty Acids n-3	4.24	6.54	6.60	1.16	0.540
n-6/n-3	3.52	2.27	2.18	0.45	0.222
Conjugated Linoleic Acid	0.40	1.05	1.07	0.33	0.401

SEM = standard error of the mean; NE = not estimable; TL = total lipids.

**Table 5 foods-11-00398-t005:** Effect of the diet on the lipid profile of skeletal muscle.

	CON	CHE	ENR	SEM	*p*-Value
Total Lipids (g/100 g of muscle)	19.25	26.84	17.15	4.73	0.252
Phospholipids (g/100 g TL)	11.25 ^b^	9.72 ^b^	20.32 ^a^	3.36	0.013
Triglycerides (g/100 g TL)	87.58 ^a^	88.11 ^a^	75.55 ^b^	3.37	0.011
Free Fatty Acids (g/100 g TL)	0.45 ^b^	3.75 ^a^	3.30 ^a^	1.37	0.033
Free Cholesterol (mg/100 g TL)	15.60 ^A^	7.84 ^B^	7.53 ^B^	1.36	<0.001
Esterified Cholesterol (mg/100 g TL)	2.50	4.73	3.05	1.22	0.456
Saturated Fatty Acids	41.19 ^C^	63.52 ^A^	55.46 ^B^	3.15	0.006
Monounsaturated Fatty Acids	38.27 ^A^	26.03 ^C^	29.75 ^B^	1.35	<0.001
Polyunsaturated Fatty Acids	19.96 ^a^	11.48 ^b^	17.35 ^a^	2,22	0.019
Polyunsaturated Fatty Acids n-6	18.39 ^a^	9.47 ^c^	11.48 ^b^	1,39	0.003
Polyunsaturated Fatty Acids n-3	1.60 ^c^	1.99 ^b^	5.34 ^a^	1.18	0.016
n-6/n-3	11.69 ^A^	5.22 ^B^	2.78 ^C^	1.03	<0.001
Conjugated Linoleic Acid	0.07 ^C^	0.42 ^B^	1.37 ^A^	0.09	<0.001

SEM = standard error of the mean. ^a,b,c^ Differences on the row *p* < 0.05. ^A,B,C^ Differences on the row *p* < 0.01.

**Table 6 foods-11-00398-t006:** Effect of the diet on the lipid profile of adipose tissue.

	CON	CHE	ENR	SEM	*p*-Value
Total Lipids (g/100 g of ad. tissue)	64.17	51.62	57.76	4.10	0.175
Phospholipids (g/100 g TL)	1.11	1.12	1.02	0.25	0.641
Triglycerides (g/100 g TL)	95.03	95.03	95.00	0.23	0.953
Free Fatty Acids (g/100 g TL)	2.27	1.61	1.81	0.27	0.611
Free Cholesterol (mg/100 g TL)	5.60	7.84	7.53	1.36	0.584
Esterified Cholesterol (mg/100 g TL)	2.50	4.73	3.05	1.22	0.430
Saturated Fatty Acids	48.92 ^B^	55.16 ^A^	53.08 ^A^	1.45	<0.001
Monounsaturated Fatty Acids	27.91 ^B^	34.54 ^A^	36.35 ^A^	0.58	<0.001
Polyunsaturated Fatty Acids	22.17 ^A^	9.30 ^B^	9.57 ^B^	0.61	<0.001
Polyunsaturated Fatty Acids n-6	21.14 ^A^	7.67 ^B^	6.57 ^B^	0.56	<0.001
Polyunsaturated Fatty Acids n-3	0.78	0.62	0.61	0.07	0.766
n-6/n-3	27.10 ^A^	12.17 ^B^	10.77 ^B^	0.65	<0.001
Conjugated Fatty Acids	0.02 ^C^	0.89 ^B^	2.31 ^A^	0.09	<0.001

SEM = standard error of the mean; TL = total lipids. ^A,B,C^ Differences on the row *p* < 0.01.

**Table 7 foods-11-00398-t007:** Effect of the diet on the gene expression of liver, brain, skeletal muscle and adipose tissue.

LIVER						BRAIN					SKELETAL MUSCLE					ADIPOSE TISSUE				
	CON	CHE	ENR	SEM	*p*-Value	CON	CHE	ENR	SEM	*p*-Value	CON	CHE	ENR	SEM	*p*-Value	CON	CHE	ENR	SEM	*p*-Value
Lipid metabolism																				
PLIN1	-	-	-	-	NE	-	-	-	-	NE	-	-	-	-	NE	0.55	0.67	0.32	0.11	0.572
PLIN2	1.27 ^a^	0.40 ^b^	0.57 ^b^	0.20	0.002	1.13 ^A^	1.13 ^A^	0.59 ^B^	0.1	0.003	1.1	0.82	0.83	0.12	0.123	1.29	0.87	0.75	0.17	0.975
PLIN3	1.29 ^a^	1.41 ^a^	0.75 ^b^	0.15	0.024	1.14	1.33	0.65	0.2	0.435	0.93 ^a^	0.45 ^b^	0.75 ^a^	0.1	0.045	-	-	-	-	NE
PLIN4	1.09 ^c^	12.34 ^b^	18.67 ^a^	2.81	0.008	2.96	2.55	2.45	0.75	0.655	1.21 ^A^	0.49 ^B^	0.43 ^B^	0.11	0.009	1.10 ^A^	0.37 ^B^	0.21 ^B^	0.09	<0.001
PLIN5	0.59 ^b^	0.36 ^b^	1.08 ^a^	0.12	0.005	2.46	1.39	1.89	0.61	0.699	0.78 ^A^	0.22 ^B^	0.20 ^B^	0.08	<0.001	-	-	-	-	NE
SREBP1	2.25 ^b^	3.38 ^a^	1.26 ^c^	0.43	0.025	1.20 ^A^	0.57 ^C^	0.89 ^B^	0.08	0.007	0.69	1.16	0.88	0.17	0.968	1.23 ^b^	5.39 ^a^	2.46 ^b^	0.87	0.029
ACACA	1.38 ^a^	0.74 ^b^	0.76 ^b^	0.16	0.037	1.52	1.31	1.03	0.24	0.213	0.78 ^A^	0.74 ^A^	0.29 ^B^	0.09	0.004	2.51	1.94	2.94	0.5	0.144
ACACB	1.73 ^A^	0.37 ^B^	0.55 ^B^	0.23	<0.001	-	-	-	-	NE	0.76	0.65	0.70	0.10	0.650	1.50 ^A^	0.42 ^B^	0.56 ^B^	0.13	0.002
FASN	1.08 ^A^	0.24 ^B^	0.09 ^C^	0.06	<0.001	1.54 ^b^	1.38 ^b^	3.54 ^a^	0.43	0.034	0.84	0.91	0.91	0.08	0.411	1.40 ^C^	6.56 ^B^	16.65 ^A^	0.9	<0.001
SCD	1.09 ^A^	0.37 ^C^	0.65 ^B^	0.07	<0.001	0.92 ^a^	1.13 ^a^	0.34 ^b^	0.15	0.021	1.69 ^A^	0.44 ^B^	0.39 ^B^	0.2	0.004	1.1	0.86	0.97	0.18	0.167
GPAM	1.14 ^A^	1.31	1.59	0.29	0.419	1.50	1.21	1.07	0.21	0.517	1.50	1.21	1.07	0.21	0.517	2.07 ^A^	0.33 ^B^	0.33 ^B^	0.23	0.003
AGPAT2	1.48	0.93	0.99	0.31	0.839	1.43	1.62	2.31	0.54	0.321	0.86	0.93	1.05	0.2	0.635	2.13	4.68	2.73	1.00	0.383
DGAT1	0.84	1.00	0.78	0.16	0.408	1.61 ^A^	1.01 ^B^	0.33 ^C^	0.16	<0.001	0.63 ^a^	0.63 ^a^	0.34 ^b^	0.11	0.034	0.99	1.10	0.91	0.21	0.901
DGAT2	1.14	1.49	1.56	0.26	0.496	1.17 ^A^	0.62 ^B^	0.32 ^B^	0.15	0.007	0.95 ^A^	0.30 ^B^	0.45 ^B^	0.06	<0.001	2.36 ^B^	6.92 ^A^	0.94 ^B^	0.88	0.002
LPIN1	2.10 ^c^	10.17 ^b^	22.27 ^a^	3.78	0.021	1.13 ^a^	0.96 ^a^	0.43 ^b^	0.17	0.016	1.13 ^A^	0.65 ^B^	1.36 ^A^	0.11	0.006	1.83 ^A^	0.56 ^B^	0.48 ^B^	0.13	<0.001
ELOVL2	1.16	0.75	1.18	0.2	0.658	1.62 ^a^	1.50 ^a^	0.33 ^b^	0.32	0.020	-	-	-	-	NE	-	-	-	-	NE
ACLY	1.63 ^A^	0.58 ^B^	0.24 ^C^	0.11	<0.001	1.11 ^a^	0.96 ^a^	0.63 ^b^	0.12	0.016	0.28	0.26	0.13	0.06	0.863	1.63 ^B^	3.64 ^A^	2.27 ^B^	0.21	<0.001
ETNK1	0.80	0.68	0.61	0.08	0.881	1.48 ^A^	0.59 ^B^	0.51 ^B^	0.15	0.008	0.95 ^A^	0.43 ^B^	0.31 ^B^	0.06	<0.001	2.58	2.48	1.26	0.37	0.688
PCYT2	1.00 ^A^	0.83 ^B^	0.34 ^C^	0.07	<0.001	1.10 ^A^	0.67 ^B^	0.32 ^C^	0.1	0.007	1.28 ^B^	2.31 ^A^	1.15 ^B^	0.17	0.001	1.49	2.7	2.85	0.43	0.975
CEPT-1	0.57	0.35	0.54	0.09	0.754	1.40	1.44	1.35	0.35	0.445	0.84 ^A^	1.06 ^A^	0.31 ^B^	0.09	<0.001	1.59	1.51	1.06	0.2	0.916
PISD	0.98 ^b^	1.03 ^b^	1.56 ^b^	0.13	0.035	1.36	1.64	1.54	0.23	0.644	0.79 ^a^	0.45 ^b^	0.76 ^a^	0.09	0.045	2.77	1.03	1.32	0.57	0.732
CERK	1.71	1.63	1.17	0.32	0.761	1.23	0.99	0.6	0.14	0.029	0.82 ^A^	0.41 ^B^	0.25 ^B^	0.06	0.002	1.94 ^A^	1.15 ^B^	0.67 ^C^	0.14	0.001
CERS6	2.13	1.88	2.54	0.52	0.384	1.37	0.91	1.14	0.23	0.714	-	-	-	-	NE	2.02	1.60	1.12	0.21	0.202
PCK1	0.92 ^b^	0.66 ^c^	1.29 ^a^	0.17	0.029	-	-	-	-	NE	-	-	-	-	NE	0.92 ^A^	0.22 ^B^	0.16 ^B^	0.1	0.002
PRKAA1	0.76 ^b^	1.10 ^a^	0.82 ^b^	0.05	0.006	1.38	0.91	1.46	0.24	0.816	0.44	0.40	0.41	0.05	0.444	2.67 ^a^	1.29 ^b^	0.98 ^b^	0.32	0.027
ATGL	1.33	1.78	1.65	0.27	0.385	1.78	1.78	1.5	0.35	0.885	0.79	0.43	0.58	0.11	0.738	1.61 ^A^	0.24 ^B^	0.26 ^B^	0.15	<0.001
CPT1 b	1.03 ^b^	1.26 ^b^	2.45 ^a^	0.41	0.036	-	-	-	-	NE	0.94	0.68	0.99	0.10	0.489	2.37 ^A^	1.97 ^A^	0.00 ^B^	0.6	0.007
LIPE	1.58	2.44	1.93	0.28	0.843	1.54	1.32	0.91	0.28	0.421	0.45	0.42	0.31	0.07	0.521	1.86	0.94	0.91	0.31	0.641
Inflammation and cell cycle																				
PRKCQ	0.52 ^A^	0.45 ^A^	0.28 ^A^	0.14	0.005	1.42 ^A^	1.11 ^B^	0.32 ^C^	0.14	<0.001	0.98 ^A^	1.24 ^A^	0.36 ^B^	0.13	0.009	-	-	-	-	NE
NFATC2	-	-	-	-	NE	-	-	-	-	NE	1.11 ^B^	7.26 ^A^	2.46 ^B^	0.52	<0.001	1.56	1.36	1.39	0.29	0.669
TRP53	1.12 ^a^	0.51 ^b^	0.24 ^c^	0.12	0.002	1.46 ^A^	0.77 ^B^	0.25 ^C^	0.15	0.006	1.18 ^b^	2.01 ^a^	1.13 ^b^	0.2	0.011	1.08 ^a^	1.12 ^a^	0.29 ^b^	0.18	0.012
P21	1.91 ^A^	1.59 ^A^	1.84 ^A^	0.27	0.194	1.05	0.88	0.9	0.16	0.589	0.71 ^A^	0.23 ^B^	0.28 ^B^	0.07	0.002	1.54 ^A^	0.29 ^B^	0.32 ^B^	0.14	<0.001
FGF21	0.28 ^b^	0.08 ^c^	0.43 ^a^	0.1	0.024	-	-	-	-	NE	-	-	-	-	NE	0.82 ^A^	0.10 ^B^	0.15 ^B^	0.04	<0.001
IL1 B	1.11 ^A^	0.33 ^B^	0.41 ^B^	0.10	<0.001	-	-	-	-	NE	-	-	-	-	NE	0.25	0.43	0.00	0.06	0.530
Mitochondrial metabolism																				
PGC1 A	1.21 ^c^	1.90 ^b^	2.20 ^a^	0.18	0.002	1.36	0.96	1.62	0.23	0.662	1.36	0.96	1.62	0.23	0.662	3.15	2.33	2.05	0.28	0.535
OPA-1	0.77 ^c^	1.22 ^b^	1.60 ^a^	0.19	0.022	1.14	0.83	0.85	0.11	0.435	1.14	0.83	0.85	0.11	0.435	1.69	1.92	1.42	0.23	0.622
MFN1	0.83 ^a^	0.40 ^b^	0.35 ^b^	0.10	0.043	-	-	-	-	NE	-	-	-	-	NE	2.67	7.98	4.69	1.88	0.789
DRP1	1.16	0.95	0.84	0.12	0.654	-	-	-	-	NE	-	-	-	-	NE	1.99	2.26	1.94	0.26	0.964
UCP2	1.71 ^a^	0.74 ^c^	1.14 ^b^	0.26	0.014	1.38 ^A^	0.55 ^B^	0.38 ^B^	0.13	<0.001	1.38 ^A^	0.55 ^B^	0.38 ^B^	0.13	<0.001	1.80 ^A^	0.60 ^B^	0.69 ^B^	0.15	<0.001
UCP3	-	-	-	-	NE	-	-	-	-	NE	-	-	-	-	NE	1.27	0.82	1.00	0.15	0.720
NDUFS3	0.97 ^A^	0.99	1.71	0.36	0.791	0.99	1.45	1.24	0.18	0.954	0.99	1.45	1.24	0.18	0.954	1.51 ^A^	1.06 ^A^	0.34 ^B^	0.17	0.001
UQCR10	0.67 ^A^	0.55	0.66	0.13	0.756	2.09 ^a^	1.10 ^b^	1.03 ^b^	0.27	0.013	2.09 ^a^	1.10 ^b^	1.03 ^b^	0.27	0.019	3.72 ^A^	2.31 ^B^	0.17 ^C^	0.55	<0.001
COX10	1.77 ^a^	1.34 ^b^	1.04 ^c^	0.16	0.034	1.54	1.78	1.14	0.31	0.484	1.54	1.78	1.14	0.31	0.484	1.74 ^A^	1.79 ^A^	0.81 ^B^	0.15	0.007
ATP5 G1	0.81 ^a^	0.68 ^b^	0.46 ^c^	0.10	0.016	1.56 ^a^	0.72 ^b^	0.92 ^b^	0.24	0.022	1.56 ^a^	0.72 ^b^	0.92 ^b^	0.24	0.026	2.26	4.67	3.75	0.87	0.675
KLOTOB	-	-	-	-	NE	-	-	-	-	NE	-	-	-	-	NE	0.96 ^A^	0.09 ^B^	0.19 ^B^	0.08	<0.001

SEM = standard error of the Mean; NE = not estimable. ^a,b,c^ Differences on the row *p* < 0.05. ^A,B,C^ Differences on the row *p* < 0.01. The acronyms of the genes are explained in Appendix A.

## Data Availability

The datasets analyzed during the current study are available from the corresponding author on reasonable request.

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
