# Peer review of "Supplementation of Enriched Polyunsaturated Fatty Acids and CLA Cheese on High Fat Diet: Effects on Lipid Metabolism and Fat Profile"

_foods, 2022, doi:10.3390/foods11030398_

Round 1

Reviewer 1 Report

The authors investigated n-3 PUFA- and CLA-riched cheese on its in-vivo lipid improvement effects. The following issues should be solved:

  1. The fact that n-3 PUFA and CLA can improve lipid metabolism is not new. There have been reports on their beneficial effects. So, the authors should evidently claim the innovative highlights.
  2. Animal grouping: there is no model used in this work. Therefore, by only comparing with the normal group, the authors may not be able to show the evidence sufficiently.
  3. How did the authors determine the components of the CHE and the ENR, especially for n-3 PUFA and CLA? The composition of the two diets looks very similar. So, it may be better to distinguish between the two diets in detail, and connect the effect differences to the diet composition differences.
  4. On the other hand, there are quite differences between the CON and cheese diets, especially for carbohydrates and energy. I wondered if the beneficial effects come from the major characteristic differences but not n-3 PUFA or CLA.
  5. The internal standard for GC-FID: considering that cheese is a milk product, C19:0 may appear in the samples. Would the author prove the specificity of the internal standards, e.g., by showing the compared chromatograms?
  6. Since the authors tested the expression for a lot of genes, the authors are recommended to summarize the potential pathway (such as drawing a pathway map).

Author Response

Dear Editor,

we have carefully read the email you sent us and the reviewers' comments on our manuscript. The reviewers' comments and suggestions gave us a chance to improve the work. All changes made to the manuscript are highlighted in red.

Thanking you once again for the opportunity given to us, and also thanking the reviewers for the valuable insights they gave us with their comments,

Yours faithfully

Giuseppe Conte

The authors investigated n-3 PUFA- and CLA-riched cheese on its in-vivo lipid improvement effects. The following issues should be solved:

  1. The fact that n-3 PUFA and CLA can improve lipid metabolism is not new. There have been reports on their beneficial effects. So, the authors should evidently claim the innovative highlights.

 AU: We agree with the reviewer. The positive effects of n-3 PUFA and CLA are undoubtedly known, as demonstrated by the extensive literature. However, little is still known about the possible effects of these fatty acids on various tissues, particularly in HFD with cheese. Moreover, we used cheese enriched in ALA and CLA instead of a supplement. This approach was preferred to a simple addition of w-3 mixed in the pellet as the added cheese is a dairy product already marketed and this approach should have allowed to mimic at least in part the consumption of cheese within a diet as varied as that of the consumer These considerations have been added in the text. (lines 96-99 in the revised manuscript).

  1. Animal grouping: there is no model used in this work. Therefore, by only comparing with the normal group, the authors may not be able to show the evidence sufficiently.

 AU: In this work we used a one-way ANOVA model that analyses the fixed effect of the diet, by comparing the three diets. We have tried to clarify this aspect in the text (lines 194-196 in the revised manuscript).

  1. How did the authors determine the components of the CHE and the ENR, especially for n-3 PUFA and CLA? The composition of the two diets looks very similar. So, it may be better to distinguish between the two diets in detail, and connect the effect differences to the diet composition differences.

 AU: Thank to the reviewer for this question. Protein, lipid, carbohydrates and ash contents were determined with official AOAC methods, while the fatty acid profile was determined by gas-chromatographic analysis. Diets CHE and ENR were characterized by a high content of lipids and a lower level of carbohydrates. Moreover, CHE and ENR diets showed a higher amount of C16:0, C18:0, C18:1c9 and CLA (C18:2c9t11). The comparison between the two diets with cheese shows that the ENR diet is richer in CLA and ALA, the levels of which exceed the thresholds (0.3 and 0.7% respectively) of consumer well-being, as indicated by Mele et al. (2011). The CHE diet, on the other hand, is unable to guarantee these levels, therefore it does not express the nutraceutical effects of these fatty acids. These considerations have been added in the text. (lines 108-124 in the revised manuscript)

  1. On the other hand, there are quite differences between the CON and cheese diets, especially for carbohydrates and energy. I wondered if the beneficial effects come from the major characteristic differences but not n-3 PUFA or CLA.

 AU: The goal of our work was precisely to verify if in HFD (control diet has only 4% of lipids against 23% of CHE and ENR, see table 1) the supplementation of CLA and ALA could have some effects. To understand if the effect was linked to these two components, we compared two diets with a cheese enriched in these fatty acids (ENR diet) and a cheese poor in the same acids (CHE diet). Therefore, the differences observed between CHE and ENR can be attributed to CLA and ALA as they are the two components that differ between the two diets. These considerations have been added in the text. (lines 108-124 in the revised manuscript)

  1. The internal standard for GC-FID: considering that cheese is a milk product, C19:0 may appear in the samples. Would the author prove the specificity of the internal standards, e.g., by showing the compared chromatograms?

 AU: C19:0 is a fatty acid not present in dairy products, therefore it is an optimal internal standard. This fatty acid has been used extensively an internal standard in numerous papers by us and other authors : Schreiner M. (2005, Quantification of long chain polyunsaturated fatty acids by gas chromatography: Evaluation of factors affecting accuracy, Journal ofChromatography A); Moltò-Puigmartì et al. (2007, Conjugated linoleic acid determination in human milk by fast-gas chromatography, Journal of Chromatography A); Martinez et al. (2010, Development of a simple method for the quantitative determination of fatty acids in milk with special emphasis on long-chain fatty acids, CyTA Journal of Food) ect.

  1. Since the authors tested the expression for a lot of genes, the authors are recommended to summarize the potential pathway (such as drawing a pathway map).

AU: Done as suggested by the reviewer. Thank you. (lines 308-310 in the revised manuscript)

Reviewer 2 Report

The manuscript “Supplementation of enriched polyunsaturated fatty acids and CLA cheese on High Fat Diet: effects on lipid metabolism and fat profile” is generally very well written and contains data of some relevance for a general readers as well as of high relevance for specialists in the topic.

The paper is extremely interesting. I read it several times and I will probably come back to it again. In my opinion, the research is well planned, carried out and described. From the editorial point of view, I also did not find any glaring errors.

Author Response

Dear Editor,

we have carefully read the email you sent us and the reviewers' comments on our manuscript. The reviewers' comments and suggestions gave us a chance to improve the work. All changes made to the manuscript are highlighted in red.

Thanking you once again for the opportunity given to us, and also thanking the reviewers for the valuable insights they gave us with their comments,

Yours faithfully

Giuseppe Conte

Reviewer_2

The manuscript “Supplementation of enriched polyunsaturated fatty acids and CLA cheese on High Fat Diet: effects on lipid metabolism and fat profile” is generally very well written and contains data of some relevance for a general readers as well as of high relevance for specialists in the topic.

The paper is extremely interesting. I read it several times and I will probably come back to it again. In my opinion, the research is well planned, carried out and described. From the editorial point of view, I also did not find any glaring errors. In my opinion, the paper is suitable for publication in its current form.

AU: Thank you very much

Reviewer 3 Report

Dear authors, it seems to me that this manuscript has great relevance in the scientific world. However, some few points could affect the quality of the manuscript.

Line 98: What is the experimental design in statistic terms? Add this information.

Line 128: What type of sample? In nature, dry, or cold-dry samples? Add this information.

Line 138: What is the code of tis commercial standard? Add this information.

Lines 154-156: Please, describe gases and column data used.

Lines 386-388: Remove these lines.

Lines 405-406: Remove these lines.

Lines 406-409: This text belongs to the results topic.

Lines 414-416: Do you think the stereospecificity of fatty acids is related? Study this hypothesis.

Lines 386-387: PLIN3 and PLIN5. The text refers to “metabolism” and “lipid oxidation”, respectively. Please be more specific. Is it about anabolism and catabolism, respectively?

Conclusion: the conclusion is very long. Rewrite it being more specific. Actually, the conclusion is currently more of a mix of review and abstract.

Tables: Add the actual P-value in the tables instead of the word "NS" or asterisks. Also, add the standard error of the mean SEM” instead of the “standard error”.

Table 2: You can use the expression “initial body weight” and “final body weight” instead of “Body Weight at the start of the treatment” and “body weight after 8 weeks of treatments”, respectively.Also, describe these names in the table and not in the footnote.

Table 3: if possible, describe the full name of the elements to decrease the size of the footnote.Here and in all tables.

Author Response

Dear Editor,

we have carefully read the email you sent us and the reviewers' comments on our manuscript. The reviewers' comments and suggestions gave us a chance to improve the work. All changes made to the manuscript are highlighted in red.

Thanking you once again for the opportunity given to us, and also thanking the reviewers for the valuable insights they gave us with their comments,

Yours faithfully

Giuseppe Conte

Reviewer_3

Dear authors, it seems to me that this manuscript has great relevance in the scientific world. However, some few points could affect the quality of the manuscript.

Line 98: What is the experimental design in statistic terms? Add this information.

 AU: We added the information about the experimental design. (lines 194-196 in the revised manuscript)

Line 128: What type of sample? In nature, dry, or cold-dry samples? Add this information.

 AU: Done, we added these details as suggested by the reviewer. Thank you. (line 146 in the revised manuscript)

Line 138: What is the code of tis commercial standard? Add this information.

 AU: Done, we added code as suggested by the reviewer. Thank you. (lines 157-158 in the revised manuscript)

Lines 154-156: Please, describe gases and column data used.

 AU: Done, we added informations as suggested by the reviewer. Thank you. (lines 174-177 in the revised manuscript)

Lines 386-388: Remove these lines.

 AU: Done. Thank you.

Lines 405-406: Remove these lines.

 AU: Done. Thank you.

Lines 406-409: This text belongs to the results topic.

 AU: We removed it from the discussion. Thanks to the reviewer for this suggestion. Thank you.

Lines 414-416: Do you think the stereospecificity of fatty acids is related? Study this hypothesis.

 AU: The phenomenon has not yet been clarified. We thank the reviewer for the suggestion, this topic will definitely be considered in the future. (lines 418-419 in the revised manuscript)

Lines 386-387: PLIN3 and PLIN5. The text refers to “metabolism” and “lipid oxidation”, respectively. Please be more specific. Is it about anabolism and catabolism, respectively?

 AU: Thanks to the reviewer for this comment. We better clarify this aspect, please see line …. In particular PLIN3 and PLIN5 are both involved in lipolysis. As regard our results, it is possible that the lower expression of PLIN5 may be compensated by an increase of PLIN3 expression. (lines 489-490 in the revised manuscript)

Conclusion: the conclusion is very long. Rewrite it being more specific. Actually, the conclusion is currently more of a mix of review and abstract.

 AU: Thanks to the reviewer. We rewrote the conclusion as suggested. (lines 650-651) in the revised manuscript)

Tables: Add the actual P-value in the tables instead of the word "NS" or asterisks. Also, add the “standard error of the mean SEM” instead of the “standard error”.

 AU: Done as suggested by the reviewer. Thank you.

Table 2: You can use the expression “initial body weight” and “final body weight” instead of “Body Weight at the start of the treatment” and “body weight after 8 weeks of treatments”, respectively.Also, describe these names in the table and not in the footnote.

 AU: Done as suggested by the reviewer. Thank you.

Table 3: if possible, describe the full name of the elements to decrease the size of the footnote.Here and in all tables.

AU: Done as suggested by the reviewer. Thank you..

Round 2

Reviewer 1 Report

The authors have modified the manuscript to make it of higher quality. All the questions have been answered, and the main text has been improved. I think there is no reason to decline this work.